# Relationship between popularity and the likely efficacy: an observational study based on a random selection on top-ranked physical activity apps

Paulina Bondaronek [iD],[1] April Slee,[2] Fiona L Hamilton,[1] Elizabeth Murray [iD][1]

[1]eHealth Unit, Research Department of Primary Care and Population Health, University College London, London, UK
[2]Research Department of Primary Care and Population Health, University College London, London, UK

**Correspondence to**
Dr Paulina Bondaronek;
p.bondaronek@ucl.ac.uk

## ABSTRACT

**Objectives** To explore the relationship between popularity of mobile application (apps) for physical activity (PA) and their likely efficacy. The primary objective was to assess the association between app popularity (indicated by user ratings) and likely efficacy (indicated by the number of Behaviour Change Techniques (BCT) present). The secondary objective was to assess the relationship between user ratings and those BCTs that have been shown to be effective in increasing PA.

**Design** Observational study.

**Methods** 400 top-ranked free and paid apps from iTunes and Google Play stores were screened, and were included if the primary behaviour targeted was PA and they had stand-alone functionality. The outcome variable of user rating was dichotomised into high (4, 5 stars) or low (1, 2, 3 stars) rating.

**Setting** iTunes and Google Play app stores.

**Participants** No individual participants but the study used user-led rating system in the app store.

**Primary and secondary outcome measures** BCTs and user rating.

**Results** Of 400 apps, 156 were eligible and 65 were randomly selected, downloaded and assessed by two reviewers. There was no relationship overall between star ratings and the number of BCTs present, nor between star ratings and the presence of BCTs known to be effective in increasing PA. App store was strongly associated with star ratings, with lower likelihood of finding 4 or 5 stars in iTunes compared with Google Play (OR 0.74, 95% CI 0.73 to 0.76, p<0.001).

**Conclusions** The findings of this study suggest that popularity does not necessarily imply the likelihood of effectiveness. Hence, public health impact is unlikely to be achieved by allowing market forces to 'prescribe' what is used by the public.

## Strengths and limitations of this study

► First study to assess the relationship between popularity of physical activity apps and their likely efficacy in two major app stores.
► A systematic approach to sample identification and assessment including a standardised assessment tool used to characterise behaviour change interventions.
► Sample identified and assessed by two independent reviewers.
► It is unknown what variables comprise the ranking algorithm from which the sample was derived.
► It is possible that user ratings, as expressed by the stars assigned to the apps, can relate to different aspects of the app content and functionalities.

behaviours. In 2015, 34% of mobile phone owners had at least one health-related app downloaded on their mobile phone.[2] Yet, 28.7% of adults in England are inactive.[3] This discrepancy may be explained by the difference between the intention to increase physical activity (PA) and the actual engagement in the behaviour, that is, the intention-behaviour gap.[4]

Researchers assessing the relationship between the popularity of apps and their quality have found mixed results. Azar *et al*[5] found that apps for weight management that were of higher quality, defined in their study as inclusion of the constructs from four behaviour change theories, were not among the highest ranked apps in the app stores. Similarly, apps that had higher download rates or higher ranking were associated with less adherence to guidelines in smoking cessation apps.[6] On the other hand, Pereira-Azevedo *et al*[7] reviewed the descriptions of 129 urology apps in the Google Play app store and found that higher download rates were associated with expert involvement in the development of the apps. These studies targeted different

## INTRODUCTION

The accessibility, convenience and wide reach of apps create new avenues for health behaviour change on a large scale. Out of the total 325 000 health apps available on the market in 2017,[1] the largest app groups were fitness apps. The rapidly increased market supply of the apps reflects public demand for the new means of engaging in health

health conditions and behaviours from PA, the subject of this study, and used different definitions of quality, such as consistency with behaviour change theory, expert involvement and adherence to guidelines, hence their findings need to be interpreted with caution. None of these studies used behaviour change theory to systematically assess the content of the apps in terms of their likely efficacy and how it relates to app popularity.

The Behaviour Change Technique Taxonomy (BCT-Taxonomy)[8] is a systematic method used to specify the content of behaviour change interventions.[9] The Taxonomy has also been used to quantify the inclusion of behaviour change theory in interventions, including apps.[10–12] For example, Bardus et al[10] extracted BCTs from 23 apps aimed at weight management. They used the Mobile App Rating Scale (MARS) to assess the quality of apps[13] and showed a positive association between the number of BCTs and the MARS subsets: engagement, functionality and aesthetics, as well as the overall quality of app score. This may suggest that the inclusion of BCTs may be related to the quality of the apps, as assessed by MARS.

However, the same authors found no association between user ratings and either BCTs or the quality indicators on the MARS suggesting that apps that were highly rated were not of high quality. Similarly, Crane et al[11] showed that only one BCT, 'prompt review of goals', was associated with user ratings in alcohol reduction apps. The authors concluded that there was little association between the mention of theory, BCT inclusion and popularity of the alcohol reduction apps. These findings indicate that further work is needed to explore whether popular apps are those that are of high quality and are likely to be effective. This has relevance for public health policy as the combination of popularity with likely efficacy would suggest that apps have a potential role in promoting public health, whereas a disconnect between popularity and likelihood of efficacy would raise questions about leaving the market to guide user choices in app downloads.

Systematic reviews and meta-analyses have shown that self-regulation techniques, such as goal setting, monitoring and feedback, are effective in increasing PA[14–17] behaviour and they may have cumulative effects. For example, Michie et al[15] showed that self-monitoring with other self-regulatory BCTs was more effective in increasing PA than using one of those BCTs in isolation. Self-regulation has been acknowledged as important constructs in behaviour change theories, for example, in control theory[18] and self-regulation theory.[19] Hence, the presence of these BCTs can be used as an indicator for quality of those apps and a proxy measure of their likely efficacy.

## Aim and objectives

The aim of this study was to explore the relationship between the user ratings as a marker of popularity of apps and the inclusion of BCTs as a marker of likely efficacy in

publicly available PA apps. Specifically, the primary objective was to assess the association between user ratings and the number of BCTs included in the sample of popular PA apps. The secondary objective assessed the inclusion of BCTs shown to be effective in increasing PA behaviour, in particular BCTs related to self-regulation of behaviour.

## METHODS

### Design

The study used a random sample of popular apps to determine the association between user ratings and the presence of BCTs. Descriptive data included the cost and size of the app; the number and distribution of star ratings; and the presence of BCTs.

### Data sources and collection

The sample included popular PA apps identified from 400 top-ranked free and paid apps from the Health and Fitness category (100 iTunes free+100 iTunes paid+100 Google Play free+100 Google Play paid) available in the UK version of iTunes and Google Play app stores on 17 October 2016. Apps were included if (1) the primary behaviour targeted was PA; and (2) they had stand-alone functionality. Apps that specifically targeted children were excluded. These apps had to include phrases that suggested clear targeting of children. For example, 'Yoga for kids', 'Workout for kids', and 'Fun fitness for kids', 'Toddler activities' would have been excluded. The rationale for excluding apps aimed at children was (1) children may not have access to iTunes or Google Play accounts; (2) ratings may reflect the parental ratings rather than the children's; and (3) the determinants of PA in children differ from those for adults, with family and school-based activities having a major influence.[20 21] However, such apps were not included in the 400 top-ranked apps. Two reviewers (PB, Ghadah Alkhaldi) reviewed each app, assessed whether specific BCTs were present and extracted relevant data. If the apps existed in both stores, then the reviewers only downloaded and assessed apps in iTunes. The detailed process of app identification and sample description is provided in ref 22. Based on the number of ratings, the proportion and the distribution of star ratings we were able to reconstitute individual-level response data within each store.

### Data extraction

#### BCT extraction as a measure of likely efficacy

The presence or absence of 93 BCTs in the BCT-Taxonomy v1 as described by Michie et al[8] was classified using three categories: absent, appears to be present but evidence is not clear (+), present beyond doubt and evidence clear (++). The presence of self-regulation BCTs, associated with PA intervention effectiveness,[14–16 16 17] was classified using the BCT-Taxonomy grouping 1: goal and planning, and grouping 2: feedback and monitoring, as these groups reflect the self-regulation BCTs in the Taxonomy.

### User ratings

We extracted the star ratings (1–5 stars) assigned to the apps in both stores and the number of ratings assigned to each app. The average star rating was calculated by summing the number of stars awarded across all users and dividing them by the number of users that submitted the rating. When an app appeared in both stores, a weighted average of the star ratings for each store was calculated based on the relative proportion of the ratings in each store. This algorithm is equivalent to summing the number of stars awarded by all users in both stores and then dividing them by the total number of reviews across both stores. This calculation weights users equally regardless of the platform used to access the app. The variable was dichotomised into high (4, 5 stars) or low (1, 2, 3 stars) rating.

Variables determined a priori potentially to affect the relationship between BCTs and higher or lower ratings were app store (iTunes or Google Play), number of features, whether the app was free or required payment, size (in megabytes) and usability.

### Number of features

Health apps use technology-enhanced features to deliver BCTs in order to influence behaviour. The apps were categorised according to the features offered by the app, for example, PA tracking, reminders, app community, data visualisation, and so on. To the authors' best knowledge, at the time of writing, there was no standardised list of features that are commonly used in PA apps. The first 10 apps extracted were used to compile a list of PA app features. The list was continuously updated throughout the app extraction process in order to accommodate for new features that were found in the PA app sample. The features were extracted by two reviewers (PB and GA, and any discrepancies were resolved by comparing the results of the extraction and reaching consensus).

### Usability

Usability was assessed using the System Usability Scale (SUS),[23] which consists of 10 items ranked on a 5-point Likert scale, from 'strongly disagree' to 'strongly agree'. To make the wording of the scale more applicable to the study, two changes were made: (1) the wording of the eighth statement was changed from 'cumbersome' to 'awkward' as recommended[24–26]; and (2) the word 'system' was replaced by 'app'. SUS yields scores from 0 to 100 where higher scores indicate greater usability. The interpretation of the SUS score used the thresholds proposed and validated by Bangor *et al*,[24] with 72.5 described as good usability.

### Data analysis

Agreement statistics including the prevalence and bias-adjusted kappa (PABAK)[27] statistic and unadjusted kappa[28] were calculated. Disagreements were then resolved by discussions among the two reviewers and consultation with other authors if unresolved.

The number of BCTs in the apps was summarised using the mean, SD, median, 25th and 75th percentiles, and the maximum and minimum. Similar statistics were used to summarise user ratings, cost, size and SUS score. The summary descriptive tables were presented for free and paid apps separately and in total as app stores have separate rankings based on the cost.

The prespecified primary analysis was based on a linear regression of the number of BCTs on star average as this continuous outcome required the smallest sample of apps for adequate statistical power. Logistic regression to determine the relationship between the number of BCTs and the odds of high and low star ratings was the prespecified secondary analysis. We modelled a higher (vs lower) star rating as the event.

### Sample size calculations

The sample size calculation was based on a pilot sample of 10 apps (five paid from iTunes, five free from Google Play) selected from the 400 apps identified. Apps were sorted in order of store rankings. From iTunes, 38 potentially eligible paid apps were identified and every eighth app was included in the pilot sample (n=38/5=7.6=~8). From Google Play, 55 potentially eligible free apps were identified and every 11th app was included in the pilot sample (n=55/5=11). If a sample app was downloaded and found to be ineligible, the next lowest ranked app was used instead. Three apps from Google Play and none from iTunes were found to be ineligible and replaced.

A pilot study based on the 10 apps suggested that mean star ratings were normally distributed. Hence, a sample size calculation was undertaken using mean ratings as a continuous measure in a linear model. Based on this framework, a sample size of 51 apps would provide 90% power to detect a change of 0.11 for each additional BCT at 5% significance level (type I error rate). A sample size of 65 apps was selected to account for any randomly selected apps that might not fulfil the inclusion criteria once downloaded. However, the pilot sample was not sufficiently large enough to show that there was a skew in star average for the complete sample once data were extracted, which would preclude the use of linear models. The difference in star rating by store was also unknown at the time of study planning, and hence the app store interactions were a post hoc addition to this plan. A retrospective power analysis was conducted. Based on simulations, this study had 97% power to detect an OR≥1.2 with 64 apps. Since we had high power and did not see a significant result, this suggests that the true increase in the chance of a star rating of 4 or 5 with each additional BCT has an OR<1.2.

As indicated in the analysis plan, star ratings of 4 and 5 were classified as high ratings, while ratings of 1, 2 and 3 were classified as low ratings. We used the user ratings data from each store in regression models with a random effect to account for store differences. Modelling proceeded in a series of steps. First, we tested a small set of variables identified in the analysis plan for

the relationship to star rating (cost, usability, number of features, size) by including each of these as fixed covariates in univariate logistic regression models with the outcome of high versus low ratings. Any variables significant at p≤0.1 were included in models examining BCTs. App store and the app store-by-BCT interaction terms were included in each model to reflect that the clientele for iTunes and Google Play may be different, and that these differences may impact the relationship between BCTs and star ratings. Weighted logistic regression (by the number of responses for each app) was used with a random intercept term for app, reflecting that the apps selected for analysis were a random sample of available PA apps, and to control for correlation when the same app was reviewed in both stores. Analyses were performed using SAS V.9.3 and R V.3.3.3.

### Sensitivity analysis
Sensitivity analyses included a revision of the number of BCTs including only those that were present beyond any doubt (++),[29] and dichotomising user ratings into 5 stars versus <5 stars.

### Patient and public involvement
This research was conducted without patient involvement.

# RESULTS
## Sample
Out of 400 apps, 244 were excluded (see figure 1). Of the remaining 156 PA apps, 31 were duplicates in that the same app appeared in both iTunes and Google Play. Thus, 125 unique apps were eligible for random selection; 65 unique apps were randomly selected for analysis including 32 free apps and 33 paid apps. One app, Break, was excluded from analyses of user reviews as the data on user ratings were not available due to the small number

**Table 1**  Characteristics of PA apps included in the sample

|  | Free (n=32) | Paid (n=32) | Total (n=64) |
|---|---|---|---|
| **BCTs (n)** | | | |
| Mean±SD | 6.56±2.99 | 7.56±2.87 | 7.06±2.95 |
| Median | 7.00 | 8.00 | 8.00 |
| 25th, 75th percentiles | 5.00, 8.00 | 6.00, 10.00 | 5.00, 9.00 |
| **Star average*** | | | |
| Mean±SD | 4.26±0.65 | 4.35±0.45 | 4.30±0.56 |
| Median | 4.50 | 4.49 | 4.50 |
| 25th, 75th percentiles | 4.30, 4.60 | 4.29, 4.60 | 4.30, 4.60 |
| **Features (n)** | | | |
| Mean±SD | 5.94±2.17 | 5.72±1.95 | 5.8±2.05 |
| Median | 7.00 | 8.00 | 6.00 |
| 25th, 75th percentiles | 4.00, 8.00 | 5.00, 8.00 | 4.00, 8.00 |
| **Size (MB)*** | | | |
| Mean±SD | 45.53±40.85 | 61.51±47.22 | 53.52±44.53 |
| Median | 32.45 | 53.97 | 39.64 |
| 25th, 75th percentiles | 16.82, 62.27 | 28.92, 85.02 | 25.52, 79.50 |
| **Usability (SUS)** | | | |
| Mean±SD | 81.25±12.64 | 85.23±12.02 | 83.24±12.40 |
| Median | 85.00 | 87.50 | 86.25 |
| 25th, 75th percentiles | 71.88, 91.25 | 78.75, 93.75 | 75.00, 91.88 |
| **Cost (£)*** | | | |
| Mean±SD | | 3.00±2.12 | |
| Median | N/A | 2.35 | N/A |
| 25th, 75th percentiles | | 1.78, 2.99 | |

*weighted averages of iTunes and Google Play.
BCT, Behaviour Change Techniques; MB, megabyte; N/A, not applicable; PA, physical activity; SUS, System Usability Scale.

of ratings, resulting in a final sample of 64 PA apps. These apps collectively received more than 2.8 million user ratings. Individual-level data of the characteristics of each app are provided in Bondaronek et al's[22] study.

There was substantial agreement between the reviewers in the assessment of BCTs (PABAK=0.94, 95% CI 0.93 to 0.95, kappa=0.78, 95% CI 0.75 to 0.81).

Characteristics of the apps are displayed in table 1. The number of BCTs and features was approximately normally distributed, with the mean number of BCTs being 7.1 (SD=3.0) and the mean number of features being 5.8 (SD=2.1) (see online supplementary file 1 for list of PA app features and frequency of occurrence). Star ratings, size, SUS score and the cost variables were skewed. The median star rating was 4.5 (IQR 1.9–4.9) and the median SUS score was 86.3 (IQR 75.00–91.88). Among paid apps, the median cost was £2.40 (IQR 1.78–2.99).

## User ratings
In total, 2 819 469 user ratings of the 64 apps were used in the analysis. Among these, 88.5% were 4 or 5-star reviews and 11.5% were 1, 2 or 3-star reviews. Among the covariates considered for model inclusion, good usability

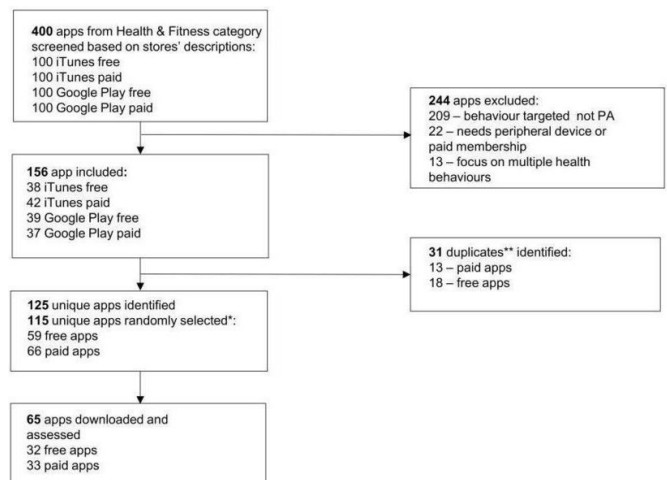

**Figure 1**  Flow chart of the apps included in the analysis. PA, physical activity. * 10 apps were assessed prior to conduct sample size calculations, hence 115 were randomly selected for the sample; ** duplicates were defined as the same app occurring in both stores.

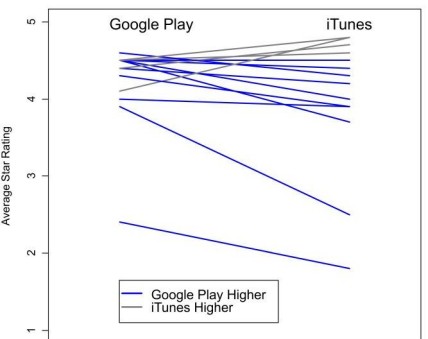

**Comparison of Star Average (n=17)**

**Figure 2** The comparison of star average between the same apps that existed in both stores (n=17).

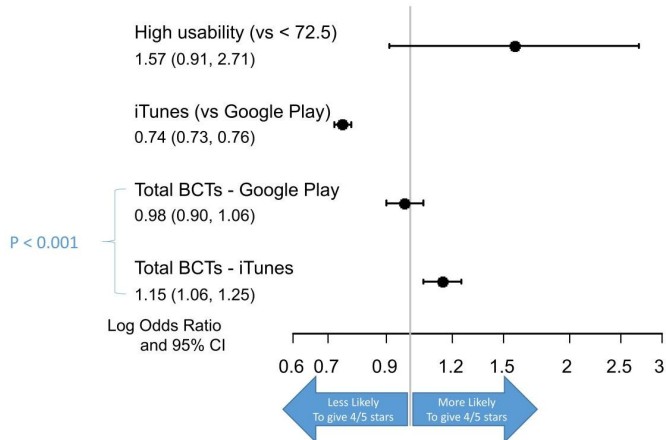

**Figure 3** ORs for the associations between the number of BCTs and a 4 or 5-star rating. BCT, Behaviour Change Techniques.

increased the chance of a 4 or 5-star rating compared with poor usability (OR 1.66, 95% CI 0.96 to 2.89, p=0.071). This covariate was included in all subsequent models. App store was strongly associated with star ratings, with lower likelihood of finding 4 or 5 stars in iTunes compared with Google Play (OR 0.74, 95% CI 0.73 to 0.76, p<0.001).

Further investigation of the difference between the stores and user ratings showed that there was 37% higher likelihood of awarding 4 or 5 stars in Google Play compared with iTunes (95% CI 35% to 41%, p<0.001). A similar pattern can be seen comparing the weighted averages within app star ratings across stores. There were 17 apps that were sold in both stores (figure 2). Analysis was conducted to assess whether the proportion with the high star ratings in Google Play exceeds what would have been found by chance. If there were no relationship between store and star rating, we would expect the star rating in iTunes to exceed the star rating in Google Play for about 50% of apps (coin flip). Mean star rating in Google Play exceeded the rating in iTunes in 13/17 apps (76.5%, 95% CI 64.2% to 86.2%, p<0.0001). Since the analysis was significant, we can rule out that the difference between the user ratings in the stores was due to chance. Cost, size and number of features were not related to star ratings.

### Primary analysis

The primary analysis of the association between user ratings and the overall number of BCTs included in each app found no relationship between number of BCTs and star rating (OR 1.05, 95% CI 0.97 to 1.14, p=0.236). Subgroup analysis showed that in iTunes only, a higher star rating was associated with the total number of BCTs (OR 0.72, 95% CI 0.71 to 0.74, p<0.001).

Based on the model containing the store interaction (figure 3), in iTunes there was an association with each additional BCT corresponding to 15% increase in the likelihood of a higher star rating (OR 1.15, 95% CI 1.06 to 1.25), there was no association between the number of BCTs and Google Play. Usability was not significant in the multivariate model (OR 1.57, 95% CI 0.91 to 2.71).

### Secondary objective

The secondary analysis examined the associations between a 4 and 5-star user rating and the inclusion of BCTs that have been shown to be effective in increasing PA behaviour.

The frequency of the BCTs associated with efficacy for increasing PA behaviour is presented in figure 4.

The BCT grouping 1 (goal and planning) was represented in 84.4% of evaluated apps. The most common of these was goal setting for behaviours (84.4%), followed by action planning (35.9%) and goal setting for outcomes (18.8%). The BCT grouping 2 (feedback and monitoring) was included in 92.2% of evaluated apps. Feedback on behaviour was included in nearly all apps (90.6%), while self-monitoring of behaviour and outcomes was incorporated in about a third of apps.

The number of grouping 1 BCTs had no impact on star rating in either store (figure 5). Among iTunes users, each additional BCT from grouping 2 increased the chance of a high star rating by 63% (OR 1.63, 95% CI 1.23 to 2.16). There was no association between the number of BCTs and star rating among Google Play users. Self-monitoring

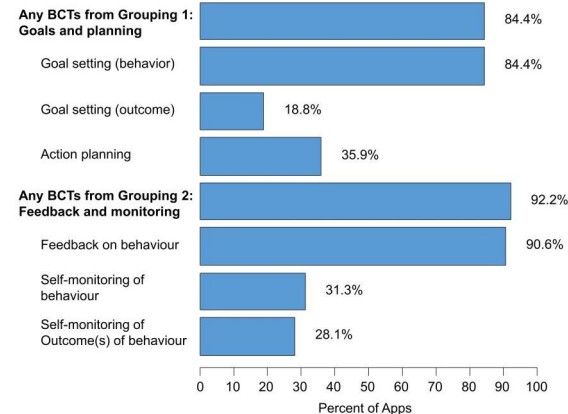

**Figure 4** Frequency of individual BCTs within the two groupings of self-regulatory BCTs. BCT, Behaviour Change Techniques.

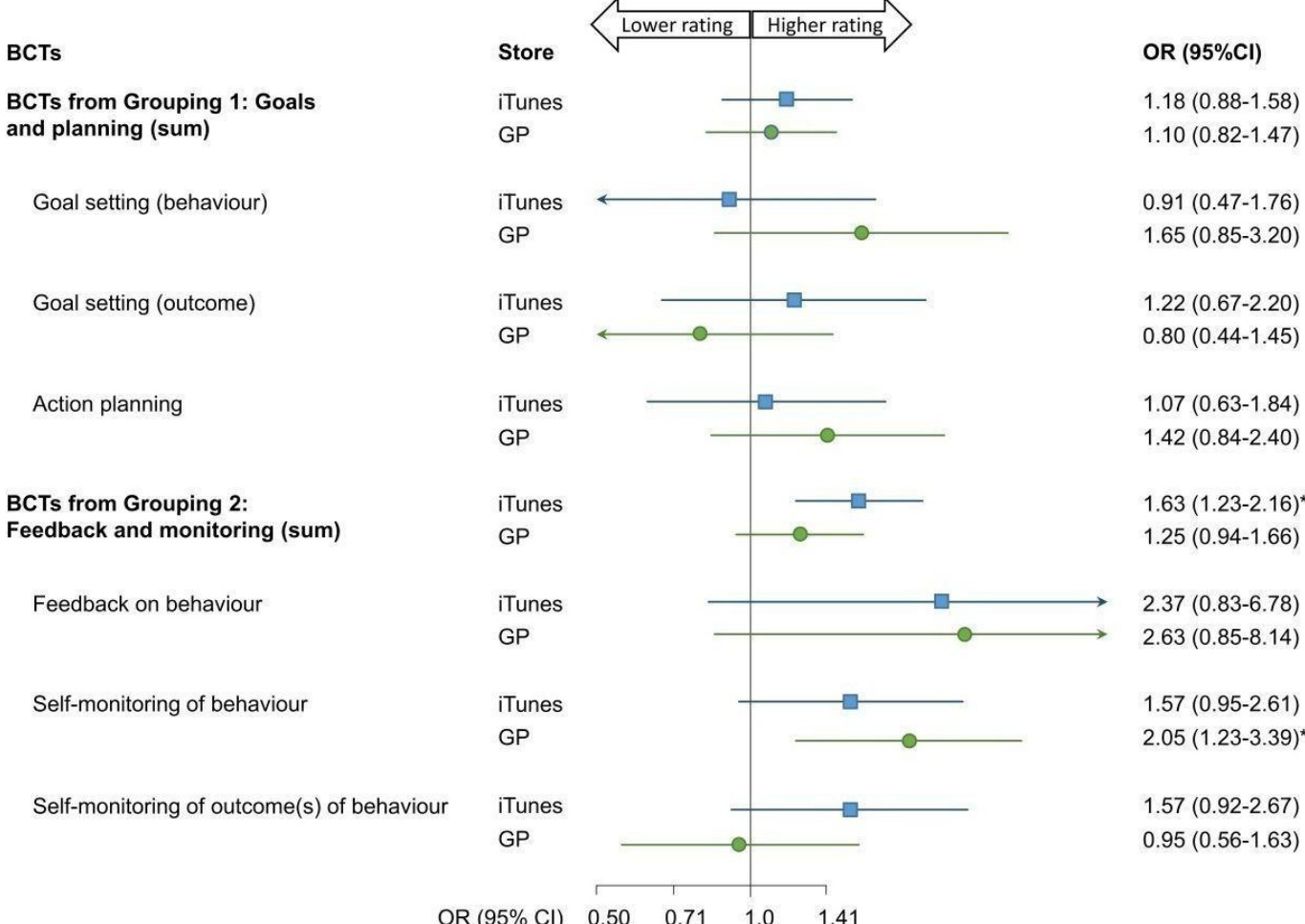

**Figure 5** ORs for the associations between BCTs that have been shown to be effective and a 4 or 5-star rating. p*< 0.05. BCT, Behaviour Change Techniques; GP, Google Play.

of behaviour was associated with higher ratings in Google Play (OR 2.05, 95% CI 1.23 to 3.39).

### Sensitivity analysis

The sensitivity analyses including only those BCTs that were classified as 'present beyond all reasonable doubt'[29] showed that for iTunes, the BCT 2.2 feedback on behaviour crossed into significance. Otherwise, the results are consistent with the original findings. The results of the sensitivity analysis are available in the online supplementary file 2.

### DISCUSSION
### Statement of principal findings

The aim of this study was to assess the relationship between the popularity of publicly available PA apps (assessed through user ratings) and their likely efficacy (assessed through the inclusion of BCTs). Overall, for both app stores, there was no association between popularity and likely efficacy as indicated by the overall presence of BCTs and the BCTs known to be effective in increasing PA. However, users in each app store differed and there

was an association between the number of BCTs and high user rating in iTunes but not in Google Play.

### Strengths and weaknesses of this study

The main strength of the study includes the systematic assessment of the app content conducted by two reviewers. The sample was identified from the most popular publicly available apps from two major app distribution platforms. The use of BCT-Taxonomy provides a standardised assessment tool, which has been used in other studies assessing the content of apps.[12 30–32] Second, Guzman[33] argued that the star rating represents an average score for the whole app that combines both positive and negative evaluations aggregated across users. The study, however, used rater-level data which included individual ratings from 2.8 million users. These large numbers mitigate the problem posed by averaging the star ratings across users.

This study has some important limitations. First, the main limitation of the study relates to the variables used in this study. It is possible that user ratings, as expressed by the stars assigned to the apps, can relate to different aspects of the app functioning and content. There is evidence suggesting that app reviews tend to occur near a

new release which can suggest that the ratings may include comments on the specific updates of the software.[34] In addition, the possibility that user ratings were influenced by fake reviews cannot be excluded.[35 36] However, the user rating was considered the most appropriate measure to explore since it represents a user-led feedback that reflects user experience. Similarly, the choice of the BCTs as an approximation for likely efficacy was selected because studies assessing the efficacy of the apps on the market are scarce.[37–39] Second, the ranking algorithm from which the sample was derived is unknown. Hence, this lack of transparency prevented evaluation of how the calculation of rank might have influenced the app selection. However, apps appear in rank order by default in app stores, hence the rank affects what users are seeing. As the aim of the study was to assess the most popular apps, the choice of highly ranked apps was considered the most appropriate for the context of this research. Third, Google Play market tends to have more ratings than iTunes because the process of app review is more complex in the latter.[40] This was addressed in the study by using the weighted averages of the ratings across the stores (for the summary of the app characteristics), and by controlling for store in the regression models. While there is a difference in the app review process in both app stores, feedback from both stores should be recognised as valid and important. Fourth, while we inspected the apps to ensure that the duplicate apps were similar in both stores, there could be differences in the functionality of the apps between the app stores that we could not see. Fifth, failure to detect the skew in the original primary outcome used to power the study is a limitation. However, a retrospective power analysis showed we had high power to detect an OR of 1.2 and did not find significant result, hence the true OR is likely to be <1.2. Lastly, the sample identification was obtained in October 2016 reflecting an extract of the state of the market at the time.

### Strengths and weaknesses in relation to other studies, discussing particularly any differences in results

This study supports the findings of previous research in apps targeting weight management,[5 10] smoking cessation[6] and alcohol reduction[11] which showed that apps that were highly rated, highly ranked or highly downloaded were not necessarily of high quality. In addition, the inclusion of BCTs that have been shown to be effective in increasing PA, that is, the self-regulation strategies, was also not associated with higher ratings and this result supports the similar findings of Bardus *et al*.[10] No association between the behaviour change theory content and user ratings suggests other factors may be contributing to the apps' popularity. High-quality graphic design, visual appeal and ease of use are more likely to attract potential customers to download and engage with the app.[41] In addition, the promotion of the apps can play a role in the download rates. The strength of this study, in comparison to other handful research exploring this association, is the use of logistic regressions, including the analysis

of the potential confounding factors, provides a strong evidence for the association, if such was existing.

The finding of an association between user rankings and overall number of BCTs in one store (iTunes) but not the other (Google Play) is new and one that has not been found in previous studies. There are various possible explanations for this finding. Market researchers suggest possible differences in the population of users of each store.[42 43] For example, iPhone users might be more affluent, they engage with their device longer, make more purchases with their phone and are more loyal to their brand.[44] In addition, there are differences in the review and approval processes between the two stores which could have influenced the user ratings.[45] Future studies should explore this observation further, and determine whether it holds for health apps targeting different behaviours.

### Meaning of the study: possible mechanisms and implications for clinicians or policymakers

In this study, we showed no evidence of association between popularity and likely efficacy. The implication of this study is that the popularity of these apps is not a sufficient filter to distinguish the apps that might have higher potential to have an impact on the potential users. Hence, we suggest that, at present, allowing the commercial market to determine which PA apps are downloaded is unlikely to be an effective method of public health promotion in terms of increasing the overall levels of PA.

Based on the findings, we suggest some implications for public health policy. Apps aimed to increase PA represent the largest category in the two major app stores[46] which illustrates public demand for engaging in PA. The lack of quality in those apps indicates a missed opportunity to increase health at the population level. Initiatives to identify and promote high-quality apps are in development, for example, the NHS Apps Library.[47] However, there is an urgent need to evaluate the effectiveness of these apps with potential users.

### Unanswered questions and future research

Further research is needed to understand which BCTs, and in which combination, are most effective in increasing PA when delivered via an app; how these BCTs can best be delivered; and how to combine features which promote efficacy with those that promote popularity. Last, the differences in iTunes and Google Play users are an unexpected finding, and not one that the study set out to identify (ie, not an a priori hypothesis). Future researchers should be aware of the potential differences between iTunes and Google Play users and ensure research is carried out on both platforms.

### CONCLUSION

To date, this is the first study to assess the association between popularity (measured using user ratings of the apps) and likely efficacy (measured using the inclusion of the BCTs) of publicly available highly ranked PA apps available in the major app stores. No relationship was

found between popularity and likely efficacy suggesting that popularity does not assure high quality, and what is liked may not be what is likely to be effective. However, PA apps in this study were highly rated, highly ranked apps from the major app stores, hence highly visible for the potential user. Hence, promotion of public health is unlikely to be achieved by allowing market forces to determine which PA apps are used. More studies are needed to assess the effectiveness of apps with users in a real-world setting to investigate the app components that are both effective and valued by the users.

**Acknowledgements** We acknowledge Ghadah Alkhaldi (Community Health Sciences Department, College of Applied Medical Sciences, King Saud University, Riyadh, Saudi Arabia) for her help with the data extraction.

**Contributors** PB designed the study, collected the data, analysed the data and wrote the manuscript. AS analysed the data and cowrote the manuscript. EM and FLH provided advice on study design, data analysis and interpretation, and contributed to the critical appraisal of the manuscript.

**Funding** PB is a PhD student funded by the Medical Research Council.

**Competing interests** None declared.

**Patient consent for publication** Not required.

**Provenance and peer review** Not commissioned; externally peer reviewed.

**Data availability statement** Data are available upon reasonable request.

**ORCID iDs**
Paulina Bondaronek http://orcid.org/0000-0003-0096-1234
Elizabeth Murray http://orcid.org/0000-0002-8932-3695

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
