## [Reviewer comments · BMJ Open]

ARTICLE DETAILS

TITLE (PROVISIONAL)	Relationship between popularity and the likely efficacy: an observational study based on a random selection on top-ranked physical activity apps
AUTHORS	Bondaronek, Paulina; Slee, April; Hamilton, Fiona; Murray, Elizabeth

VERSION 1 – REVIEW

REVIEWER	Kwok Ng University of Eastern Finland, Finland; University of Limerick, Ireland
REVIEW RETURNED	14-Mar-2019

GENERAL COMMENTS	The authors have investigated the BCTs of the top 400 apps from the iTunes and GP store with particular reference to targeting PA. Examination of the apps' popularity and the likelihood of effectiveness was carried out through logistic regressions. The authors conclude that the association was not significant and explicitly describe the public health impact. There is appropriate English language used, although the scientific language could be improved through the removal of anthropomorphisms. More comments about the study are covered in my review. Overall comments The author' introduction could be improved by matching the literature with the purpose of the study. The study design is based on a random sample of popular apps, yet in the introduction the authors have found literature that suggest that popularity and efficacy do not go hand in hand. This brings to question the methodology of the randomness, the selection from only the top 400 out of potential 325,000 apps (that the authors claim in the second sentence), and how that may influence the aims of the study. The authors do however use an intriguing way of collecting data from almost 3M users. There could be a better explanation of this process in the paper. Even if the authors feel they may have presented it in an earlier paper, a small summary would be good for the reader. I am hesitate to agree with the authors' presentation of the results of the OR. The OR is not more or less 'likely to give'. Strictly speaking, it is the likelihood to find a rating. It is not clear that the authors have used individual ratings in the procedures. If they did, can they describe how they pooled this information from the two stores. The authors have carried out much analyses that indicate the differences between the two stores, and should be included as one of the secondary aims of the study. Therefore, this should be reflected in the background literature and discussion. In simple terms, how do apple users differ from google users?
--

	In addition, the study is lacking an indepth discussion section. The author may like to consider what this actually means and how it may be interpreted for the purpose of public health. The authors have also not considered some of the behavioural theories in their discussion, let alone considered other paradigms of human-computer interaction, socialising, and so forth. Specific comments P4. L4-5, there is a bridging gap missing between fitness apps being the largest app group, and that there are almost 30% of the English public who are inactive. Are the previous statistics based solely on the English public? Why would the largest app groups be related to the proportion of inactivity? Are the authors suggesting that there is evidence that supports there would be a link? These are some assumptions that could be misinterpreted and I would like to see this reconceptualised. P4. L17, write GP out in full for the first time in the main text. P5.L15. It is unclear if the top 400 were top 400 from each app store, or 200 from each, or how this 400 was collated from the two app stores. P5.L17. A description is needed to define how this inclusion criteria was met for 'targeting adults'. Moreover, the scientific rationale for this as a criteria in the study is missing. P5.L24. avoid single line paragraphs. P5.L33. reference is missing for self-regulating BCTs that are associated with intervention effectiveness. Moreover, an explanation is needed to state why only two groups were formed. P5L44. Include a sentence from the abstract about how high and low ratings were calculated. P5.L51 the list of the features would be a really good table to see with the frequency of appearance. Please include it here. P6.L27, replace 4 or 5 star rating... with low and high rating. State which one is the reference category. P6L32, could you explain a bit more about the selection of the 10 apps in the pilot study to help illustrate how normal distribution was found, whereas the final sample was not. P7.L37. I'm not sure about the need to include all this information in the table, particularly in a sub row way. P8L38-42, and in the abstract; I do not see this defined within the aims of the study. They just seem to come out from nowhere and lack relevance to the main research aim. In all figures, avoid using GP as an abbreviation, and write in full. The main reason for this is that GP is better known in the medical world as the regular doctor (General Practitioner).
--	---

REVIEWER	Kristen DiFilippo University of Illinois, United States of America
REVIEW RETURNED	29-Mar-2019

GENERAL COMMENTS	The paper provides an important examination of if a relationship exists between the ratings of physical activity apps and the use of behavior change techniques within apps. This is important to the field when considering the impact ratings has on which apps are seen by those searching for apps. Major suggestions: Methods-Data sources and collection: It is unclear what is meant by "400 top-ranked apps." Clarification on whether the first 400 apps were pulled, or if apps were specifically searched by high star rating would make the procedures more reproducible.
--

	In the methods it states that when apps were identified in both stores, the ratings were combined; however, the results showed that the apps' ratings differed by stores. This presents a major concern. Some things to consider: does the same app always appear exactly the same in both stores? If it does not, combining is not appropriate. Also, since the ratings differ by store, it may be more appropriate to consider an app found in both stores to be two different apps. Page 8 Lines 37-49 and Page 8 Lines 52-Page 9 Line 3 present the same data, with almost identical text, except the conclusion made based on the data is opposite. Clarification is needed concerning which conclusion is correct, and to make sure the rest of the paper aligns with the correct interpretation of this data. Page 9 Line 9 the text states that there is a difference in odds ratio; however the data presented is not significant. Page 9 Line 21-22 the text states that there is not a significant difference; however the data shown is a statistically significant result. Page 10 Line 26 states that the difference between the user rankings between stores could be the play of chance; your data contradicts this conclusion. Page 10 Lines 59-60: this contradicts the description of combining apps found in both stores that is provided in the methods. Minor suggestions: Page 4 Line 8-9 include the year that this information is from. Page 4 Line 18-20: consider adding information regarding if the apps were reviewed based on the description or downloaded app. Page 5 Lines 57-59: Provide information about how many people did this, and how agreement was reached when differences occurred. Page 7: It would strengthen the paper to provide figures of bar graphs or box and whisker plots showing the skew of the data Table 1: Change N to n. Figure 1: Provide clarification on if duplicates mean that an app occurred in both stores, or more than once in a store. Page 9 Line 46: Provide the statistical test used. Page 11 Line 25: Provide a reference.
--	--

VERSION 1 – AUTHOR RESPONSE

Reviewer 1

Reviewer name: Kwok Ng

The overall comments from Reviewer 1 were extremely useful and we considered each of these carefully. This is why we divided the general comments and responded individually to the statements. Comment (C): There is appropriate English language used, although the scientific language could be improved through the removal of anthropomorphisms. More comments about the study are covered in my review.

R: Thank you for this valuable comment. We have addressed the language use throughout the manuscript.

C: The author' introduction could be improved by matching the literature with the purpose of the study. The study design is based on a random sample of popular apps, yet in the introduction the authors have found literature that suggest that popularity and efficacy do not go hand in hand. This brings to question the methodology of the randomness, the selection from only the top 400 out of potential 325,000 apps (that the authors claim in the second sentence), and how that may influence the aims of the study.

R: Thank you for this insight. We considered this comment carefully. As mentioned in the 3rd paragraph of the Introduction, research assessing the relationship between the popularity of apps and their quality has found mixed results and there is only a handful of research papers that focus on this relationship, which we described in the Introduction. As we were interested in the apps that are used (or at least downloaded) and visible to the potential users, we chose the top ranked apps. The aim of the study is to explore the association between popularity and likely efficacy, apps lower down the ranks might not be popular hence were not included in the study.

C: The authors do however use an intriguing way of collecting data from almost 3M users. There could be a better explanation of this process in the paper. Even if the authors feel they may have presented it in an earlier paper, a small summary would be good for the reader.

R: Thank you for this comment. We have added to the Methods/User rating section: "We extracted the star ratings (1-5 stars) assigned to the apps in both stores and the number of ratings assigned to each app."

C: I am hesitate to agree with the authors' presentation of the results of the OR. The OR is not more or less 'likely to give'. Strictly speaking, it is the likelihood to find a rating.

R: We fully agree with the reviewer. The language we used was imprecise and we have changed the statements to "likelihood of higher/lower user ratings".

C: It is not clear that the authors have used individual ratings in the procedures. If they did, can they describe how they pooled this information from the two stores.

R: Thank you for this comment. We have used individual ratings as, based on the number of ratings, the proportion, and the distribution of star ratings, we were able to reconstitute individual level response data within each store. We added the information based on the reviewers comments in Sections: Data source and collection, and Data analysis: We used the user ratings data from each store in regression models with a random effect to account for store differences.

C: The authors have carried out much analyses that indicate the differences between the two stores, and should be included as one of the secondary aims of the study. Therefore, this should be reflected in the background literature and discussion. In simple terms, how do apple users differ from google users?

R: Thank you for this relevant comment. We did not expect a-priori to find material differences between the stores, although fortunately our analysis plan did pre-specify a store adjustment to account for correlation. This is explained in the Data analysis/Sample size calculation section: The difference in star rating by store was also unknown at the time of study planning, and hence the app store interactions were a post-hoc addition to this plan.

We have integrated more explanation of the possible difference in iPhone and Android users to reflect the Reviewer's comment, we write: "There are various possible explanations for this finding. Market

researchers suggest possible differences in the population of users of each store 28 29. For example, iPhone users might be more affluent, they engage with their device for longer, make more purchases with their phone, and are more loyal to their brand 30. In addition, there are differences in the review and approval processes between the two stores which could have influence the user ratings 31.”

The results suggests that the ratings tended to be higher in one store for this sample of apps. We are reluctant to speculate about the reasons for these differences. It may distract from the main message for the paper hence we decided to keep the discussion to a minimum.

C: In addition, the study is lacking an indepth discussion section. The author may like to consider what this actually means and how it may be interpreted for the purpose of public health. The authors have also not considered some of the behavioural theories in their discussion, let alone considered other paradigms of human-computer interaction, socialising, and so forth.

R: The main message of the study is that the popularity might not mean quality and we focus the Discussion section on this point. We discuss the implication of this finding for public health: The findings have some implications for public health policy. Apps aimed to increase PA represent the largest category in the two major app stores 42 which illustrates public demand for engaging in PA. The lack of quality in those apps indicates a missed opportunity to increase health at the population level.

We also added a paragraph to suggest that other factors (other than evidence-based inclusion of behaviour change theory components) may influence the popularity: This suggests other factors may be contributing to the apps' popularity such as social influence, reviews from other sources, as well as the design of the apps (high quality graphic design, visual appeal and ease of use) and the specific functionalities provided. In addition, the promotion of the apps can play a role in the download rates. We have added a sentence to the Implications: “However, there is an urgent need to evaluate the effectiveness of these apps with potential users.”

C: P4. L4-5, there is a bridging gap missing between fitness apps being the largest app group, and that there are almost 30% of the English public who are inactive. Are the previous statistics based solely on the English public? Why would the largest app groups be related to the proportion of inactivity? Are the authors suggesting that there is evidence that supports there would be a link? These are some assumptions that could be misinterpreted and I would like to see this reconceptualised.

R: We agree fully with the reviewer's comment. We have incorporated into the introduction that This discrepancy may be explained by the difference between the intention to increase PA and the actual engagement in the behaviour, i.e, the intention-behaviour gap.

C: P4. L17, write GP out in full for the first time in the main text.

R: The abbreviation was changed to “Google Play” throughout the document, also reflecting your later comment suggesting that GP is more likely associated with “General Practitioner”.

C: P5.L15. It is unclear if the top 400 were top 400 from each app store, or 200 from each, or how this 400 was collated from the two app stores.

R: Thank you for this relevant comment. The section is now clarified to include the sample which was 100 iTunes free + 100 iTunes paid + 100 Google Play free + 100 Google Play paid.

C: P5.L17. A description is needed to define how this inclusion criteria was met for ‘targeting adults’. Moreover, the scientific rationale for this as a criteria in the study is missing.

R: Thank you for this feedback which is very relevant and we have changed the manuscript based on it. None of the 400 apps screened targeted children. We realised that this was an exclusion with the rationale that apps that specifically targeted children were excluded as it is likely that children might

not have access to iTunes or Google Play accounts and the ratings might reflect the parental ratings rather than the children.

Changes to the manuscript to reflect the difference: Apps that specifically targeted children were excluded as it is likely that children might not have access to iTunes or Google Play accounts and the ratings might reflect the parental ratings rather than the children.

C: P5.L24. avoid single line paragraphs.

R: We have corrected this error now and moved the sentence to the section Design.

C: P5.L33. reference is missing for self-regulating BCTs that are associated with intervention effectiveness. Moreover, an explanation is needed to state why only two groups were formed.

R: Thank you for this comment. We have added the relevant references from the Introduction section and we provided a justification why the two groups from the BCT Taxonomy (v1) were used. They were selected as these two groups reflect the self-regulation techniques.

C: P5L44. Include a sentence from the abstract about how high and low ratings were calculated.

R: We have included this sentence as suggested by the reviewer.

C: P5.L51 the list of the features would be a really good table to see with the frequency of appearance. Please include it here.

R: We agree with the reviewer. As the journal guidance is to include up to 5 figures/tables. We have included the list of the features with the frequency of appearance in the appendix instead. Please see supplementary file 2.

C: P6.L27, replace 4 or 5 star rating... with low and high rating. State which one is the reference category.

R: We have changed and added according to the reviewers suggestions.

C: P6L32, could you explain a bit more about the selection of the 10 apps in the pilot study to help illustrate how normal distribution was found, whereas the final sample was not.

R: We have added the explanation of how the 10 pilot apps were selected. The sample size calculation was based on pilot sample of 10 apps (5 paid from iTunes, 5 free from GP) selected from the 400 app identified. Apps were sorted in order of store rankings. From iTunes, 38 potentially eligible paid apps were identified and every 8th app was included in the pilot sample ($N = 38/5 = 7.6 \approx 8$). From GP, 55 potentially eligible free apps were identified and every 11th app was included in the pilot sample ($N = 55/5 = 11$). If a sample app was downloaded and found to be ineligible, the next lowest ranked app was used instead. Three apps from GP and none from iTunes were found to be ineligible and replaced.

C: P7.L37. I'm not sure about the need to include all this information in the table, particularly in a sub row way.

R: We agree with the reviewer. The table includes some redundant information. We have removed the percentiles and min, max.

C: P8L38-42, and in the abstract; I do not see this defined within the aims of the study. They just seem to come out from nowhere and lack relevance to the main research aim.

R: This is a relevant point and we addressed it in the Abstract by adding: The covariates considered for the inclusion were: number of app features, size, usability and cost.

C: In all figures, avoid using GP as an abbreviation, and write in full. The main reason for this is that GP is better known in the medical world as the regular doctor (General Practitioner).

R: The abbreviation was changed to full name "Google Play" throughout the document. Thank you for this useful point.

Reviewer: 2

Reviewer Name: Kristen DiFilippo

Institution and Country: University of Illinois, United States of America

The paper provides an important examination of if a relationship exists between the ratings of physical activity apps and the use of behavior change techniques within apps. This is important to the field when considering the impact ratings has on which apps are seen by those searching for apps.

Major suggestions:

C: Methods-Data sources and collection: It is unclear what is meant by "400 top-ranked apps." Clarification on whether the first 400 apps were pulled, or if apps were specifically searched by high star rating would make the procedures more reproducible.

R: Thank you for this relevant comment. We used top 100 iTunes free + 100 iTunes paid + 100 Google Play free + 100 Google Play paid. We considered the first 100 apps from each ranking as to focus on the apps that are most visible for the potential user. The section is now clarified and includes the sample breakdown: 100 iTunes free + 100 iTunes paid + 100 Google Play free + 100 Google Play paid.

C: In the methods it states that when apps were identified in both stores, the ratings were combined; however, the results showed that the apps' ratings differed by stores. This presents a major concern. Some things to consider: does the same app always appear exactly the same in both stores? If it does not, combining is not appropriate..

R: Thank you for this relevant concern. We downloaded 65 apps and conducted an initial inspection using both iPhone and Android phones and the apps' content was the same. We conclude that the difference in app stores is more likely to be due to the users (more specifically user ratings) and not the difference in the apps. However, we added a statement to the Limitations reflecting the Reviewers' concern: Fourth, while we inspected the apps to ensure that the duplicate apps were similar in both stores, there could be differences in the functionality of the apps between the app stores that we could not see.

C: Also, since the ratings differ by store, it may be more appropriate to consider an app found in both stores to be two different apps

R: Since the app appeared to be identical in both stores and we were concerned that the users are different, we made the decision to treat the app as the same app when appeared in the same store but adjust for store/user differences with a random effect.

C: Page 8 Lines 37-49 and Page 8 Lines 52-Page 9 Line 3 present the same data, with almost identical text, except the conclusion made based on the data is opposite. Clarification is needed concerning which conclusion is correct, and to make sure the rest of the paper aligns with the correct interpretation of this data.

R: We are grateful to the reviewer for observing this mistake. The conclusion that there is a difference between the user ratings in both stores reflects the correct interpretation. We have carefully adjusted this section removing the repetition. (April)

C: Page 9 Line 9 the text states that there is a difference in odds ratio; however the data presented is not significant.

R: Thank you again for noticing this mistake. We have corrected it now.

Relates to: The primary analysis of the association between user ratings and the overall number of BCTs included in each app found no relationship between number of BCTs and star rating (OR: 1.05, 95%CI: 0.97-1.14, $p = 0.236$). Subgroup analysis showed that in iTunes only, a higher star rating was associated with the total number of BCTs.

C: Page 9 Line 21-22 the text states that there is not a significant difference; however the data shown is a statistically significant result.

R: Thank you again for noticing this mistake. We have corrected it now.

C: Page 10 Line 26 states that the difference between the user rankings between stores could be the play of chance; your data contradicts this conclusion.

R: We agree with the reviewer. We removed this sentence.

C: Page 10 Lines 59-60: this contradicts the description of combining apps found in both stores that is provided in the methods.

R: We used the weighted averages for app characteristics (e.g., cost, size) but not in the models of user rating. This was not clear and we have clarified it now: This was addressed in the study by using the weighted averages of the ratings across the stores (for the summary of the app characteristics)

Minor suggestions:

C: Page 4 Line 8-9 include the year that this information is from.

R: Thank you. We have added this information now.

C: Page 4 Line 18-20: consider adding information regarding if the apps were reviewed based on the description or downloaded app.

R: This is a very relevant point. The review was based on the description of the apps and we have integrated this information in the Introduction now.

C: Page 5 Lines 57-59: Provide information about how many people did this, and how agreement was reached when differences occurred.

R: We have incorporated this information now. Two reviewers extracted these data.

C: Page 7: It would strengthen the paper to provide figures of bar graphs or box and whisker plots showing the skew of the data

R: We have reached the maximum number of figure and tables suggested by the journal editorial guidelines. We have attached the distribution of the star ratings which shows the pilot data and the full sample (attached file called: Distribution of star rating). Please let us know if we should include it in the supplementary files.

C: Table 1: Change N to n.

R: We agree with the reviewer. It should be “n” representing the sample of app.

C: Figure 1: Provide clarification on if duplicates mean that an app occurred in both stores, or more than once in a store.

R: We have added a definition of a duplicate (which is defined as the same app occurring in both stores) to Figure 1. It Thank you.

C: Page 9 Line 46: Provide the statistical test used.

R: The page and line relates to the frequency table, no statistical tests were used.

C: Page 11 Line 25: Provide a reference.

R: Thank you. The reference has been added.

VERSION 2 – REVIEW

REVIEWER	Kwok Ng University of Eastern Finland, Finland University of Limerick, Ireland
REVIEW RETURNED	19-Jul-2019

GENERAL COMMENTS	Round 2 review. The authors have attempted to address the comments made from the editor and both reviewers. They have made improvements to the manuscript and in some of the comments, they have responded well, but in other places, I do not feel the response is sufficient and the content in the manuscript requires further work. Specific details of this to follow below. Moreover, currently, the main limitation to the study is the extraction date of over 3 years ago. It would be a good idea for the authors to carry out the search again and conduct the analysis to reflect upon this. I do understand that this would require more work, and the authors may like to consider how to approach this
---

	when looking to publish this information. The main argument for this is that, in the last three years, there are many new apps and features that have been included, and that means the reported results from the current study may not be relevant. Although the main message may be that ratings and efficacy are not related, and this is an important message, the extent of it may have changed in the last three years. This requires investigation and unfortunately, the pace of app development is much faster than the publishing rate of science. In other journals whereby reviews of apps were included, such as JMIR, authors explicitly list the apps, yet they were not included in your study. It would be more transparent to include this list, and therefore the current supplementary file 1 would also be viewed differently in terms of which apps had which types of features. This would not be too difficult to present given that you only used 65 apps in the end. For example, in some systematic literature review of over 100 studies, authors still list the papers. Just because the apps are commercial products, it does not mean that stating them explicitly would make them more or less popular. Also, this would be an important message in terms of any conflict of interests between authors, app developers, and funders. Some of the updated comments by the authors are not necessarily substantiated. Here are some of them;  1. Children apps were excluded because children do not have access to app stores and parents would provide ratings. The authors may like to consider that children also include adolescents under the age of 18. The use of personal devices, mobile phones and so forth are actually very prevalent among children. See some of the current work by Victoria Goodyear and colleagues. Moreover, some published research that I found, demonstrated that over half of children in Finland and in Czech Republic report to have apps for physical activity promotion. I'm currently updating those study findings and we are finding there are even more young adolescents, as young as 11 years old, that own apps for PA promotion. I very much doubt this would be different in the UK, based on Goodyear and colleagues work around Birmingham. As a result, there is an inappropriate rationale for exclusion of children apps. A pertinent aspect to consider is to define and describe a criteria for what what makes an app for children or what makes it for adult when you have an exclusion criteria. 2. The discussion section is currently very weak. There were many study limitations, that the authors could have included in the discussion to reflect the results. It may be helpful for the authors to review the following paper and reconsider how to format the discussion section. https://www.bmj.com/content/318/7193/1224 3. The authors have not commented on other aspects of human computer interaction theories to help explain the findings. This may help the authors avoid the fear of speculation.
--	---

REVIEWER	Kristen DiFilippo University of Illinois at Urbana-Champaign United States of America
REVIEW RETURNED	07-Aug-2019

GENERAL COMMENTS	The paper provides a clear look into the relationship between behavior change techniques and star ratings in popular apps. For this second review, there are only minor suggestions:
--

	Remove the Intervention heading from the abstract. Table 1: It is not clear why some, but not all of the value are the "weighted average of iTunes and Google Play."
--	--

VERSION 2 – AUTHOR RESPONSE

Reviewer 1

Reviewer name: Kwok Ng

As before, the feedback section is divided so that we could respond to each of the Reviewer's valuable comments.

Comment (C) 1: It would be a good idea for the authors to carry out the search again and conduct the analysis to reflect upon this. I do understand that this would require more work, and the authors may like to consider how to approach this when looking to publish this information. The main argument for this is that, in the last three years, there are many new apps and features that have been included, and that means the reported results from the current study may not be relevant. Although the main message may be that ratings and efficacy are not related, and this is an important message, the extent of it may have changed in the last three years

Response (R): We understand the Reviewer's concerns, but effectively the Reviewer is asking us to redo the entire study from scratch. We believe that the data in this study are worthy of publication because the issue of the quality of digital health apps is an ongoing problem^{1, 2}. The study supports the rationale for the need for a "quality filter" beyond the ranking system used in the app market as the findings suggest that the popularity of an app is not an effective filter. This is important especially for the BMJ readership as health professionals are still not confident to recommend apps. However, based on the Reviewer's comment, we added a sixth limitation of the study, p.10: Lastly, the sample identification was obtained in October 2016 reflecting an extract of the state of the market at the time. We also note that Reviewer 2 has not raised this as an issue.

1 Wyatt, J. C. (2018). How can clinicians, specialty societies and others evaluate and improve the quality of apps for patient use?. *BMC medicine*, 16(1), 225.

2 Wisniewski, H., Liu, G., Henson, P., Vaidyam, A., Hajratalli, N. K., Onnela, J. P., & Torous, J. (2019). Understanding the quality, effectiveness and attributes of top-rated smartphone health apps. *Evidence-based mental health*, 22(1), 4-9.

C: In other journals whereby reviews of apps were included, such as JMIR, authors explicitly list the apps, yet they were not included in your study. It would be more transparent to include this list, and therefore the current supplementary file 1 would also be viewed differently in terms of which apps had which types of features. This would not be too difficult to present given that you only used 65 apps in the end.

R: Thank you. We agree with the Reviewer that authors explicitly list the apps in journals where app reviews were included. We have also provided an individual-level data listing all the characteristics of each app in the review and content analysis. We refer to this study in this manuscript in section: Data sources and collection, p.5: The detailed process of app identification and sample description is provided in Bondaronek, et al. 22.

However, as the Reviewer suggested, we have adapted supplementary table 1 to include the names of the apps and the number of features present. We also added to the Results section, p.7: An individual-level data of the characteristics of each app is provided in Bondaronek, et al. 22 to guide the reader if they wished to see the data per individual app.

Below are the responses to the Reviewer's numbered comments from 1-3:

C: Children apps were excluded because children do not have access to app stores and parents would provide ratings. The authors may like to consider that children also include adolescents under the age of 18. The use of personal devices, mobile phones and so forth are actually very prevalent

among children. See some of the current work by Victoria Goodyear and colleagues. Moreover, some published research that I found, demonstrated that over half of children in Finland and in Czech Republic report to have apps for physical activity promotion. I'm currently updating those study findings and we are finding there are even more young adolescents, as young as 11 years old, that own apps for PA promotion. I very much doubt this would be different in the UK, based on Goodyear and colleagues work around Birmingham. As a result, there is an inappropriate rationale for exclusion of children apps. A pertinent aspect to consider is to define and describe a criteria for what what makes an app for children or what makes it for adult when you have an exclusion criteria.

R: Thank you. The definition of the exclusion criteria was not made clear in the manuscript, as stated by the Reviewer. We defined and described the criteria for the exclusion. Please see below the changes to the manuscript.

Changes to the manuscript to reflect comment, p.5: Apps that specifically targeted children were considered for exclusion. These apps had to include phrases that suggested clear targeting of children. For example, "Yoga for kids", "Workout for kids", and "Fun fitness for kids", "Toddler activities" would have been excluded. The rationale for excluding apps aimed at children was (i) children may not have access to iTunes or Google Play accounts; (ii) ratings may reflect the parental ratings rather than the children's and (iii) the determinants of PA in children differ from those for adults, with family and school-based activities having a major influence 20 21. Such apps were not included in the 400 top ranked apps.

C: The discussion section is currently very weak. There were many study limitations, that the authors could have included in the discussion to reflect the results. It may be helpful for the authors to review the following paper and reconsider how to format the discussion section.

<https://eur01.safelinks.protection.outlook.com/?url=https%3A%2F%2Fwww.bmj.com%2Fcontent%2F318%2F7193%2F1224&data=02%7C01%7C%7Ce65f9a3f9bb44555c7cb08d72af7f622%7C1faf88fea9984c5b93c9210a11d9a5c2%7C0%7C0%7C637025116783475422&sdata=6YnueukUeFIWkxobfDa%2BLaJyvLvIIHOZZ156uxKySCs%3D&reserved=0>

R: Thank you for this comment. The overall structure as per the paper suggested by the Reviewer have been used and the Discussion has been changed to reflect the structure: a statement of the principal findings; strengths and weaknesses of the study; strengths and weaknesses in relation to other studies, discussing important differences in results; the meaning of the study: possible explanations and implications for clinicians and policymakers; and unanswered questions and future research.

We have found the structure helpful and used the headings in the manuscript. Thank you. In addition, as per Reviewer's comment, the section on limitations has been expanded.

C: The authors have not commented on other aspects of human computer interaction theories to help explain the findings. This may help the authors avoid the fear of speculation.

R: The main message of this paper is that popularity, as evidenced by user ratings, is not related to likely efficacy, as measured by presence of behaviour change techniques. While we agree that HCI theory and practice are important determinants of popularity and user ratings, we feel that discussion of HCI theories is outside of the scope of this paper.

Reviewer: 2

Reviewer Name: Kristen DiFilippo

Institution and Country: University of Illinois, United States of America

The paper provides a clear look into the relationship between behavior change techniques and star ratings in popular apps. For this second review, there are only minor suggestions:

Comment (C): Remove the Intervention heading from the abstract.

Response (R): Thank you. It has been deleted.

C: Table 1: It is not clear why some, but not all of the value are the "weighted average of iTunes and Google Play."

R: Thank you for this comment, which was helpful as it helped us to clarify the methods used. The averages for the number of BCTs, number of features, and usability were not needed because these variables were assessing the content of the apps. If the apps existed in both stores, then the reviewers only downloaded and assessed apps in iTunes. However, we realised that we this has not been described in the Methods section and has now been added, p. 5: If the apps existed in both stores, then the reviewers only downloaded and assessed apps in iTunes.